# Dual role of the foot-and-mouth disease virus 3B1 protein in the replication complex: As protein primer and as an essential component to recruit 3D$^{pol}$ to membranes

Cristina Ferrer-Orta*, Diego S. Ferrero, Nuria Verdaguer  *

Instituto de Biología Molecular de Barcelona. Consejo Superior de Investigaciones Científicas (IBMB-CSIC), Barcelona, Spain

* cfocri@ibmb.csic.es (CFO); nvmcri@ibmb.csic.es (NV)

**Data Availability Statement:** The refined coordinates and structure factors are submitted at the PDB under accession codes: 8C1N for the

## Abstract

Picornavirus genome replication takes place in specialized intracellular membrane compartments that concentrate viral RNA and proteins as well as a number of host factors that also participate in the process. The core enzyme in the replication machinery is the viral RNA-dependent RNA polymerase (RdRP) 3D$^{pol}$. Replication requires the primer protein 3B (or VPg) attached to two uridine molecules. 3B uridylylation is also catalysed by 3D$^{pol}$. Another critical interaction in picornavirus replication is that between 3D$^{pol}$ and the precursor 3AB, a membrane-binding protein responsible for the localization of 3D$^{pol}$ to the membranous compartments at which replication occurs. Unlike other picornaviruses, the animal pathogen foot-and-mouth disease virus (FMDV), encodes three non-identical copies of the 3B (3B1, 3B2, and 3B3) that could be specialized in different functions within the replication complex. Here, we have used a combination of biophysics, molecular and structural biology approaches to characterize the functional binding of FMDV 3B1 to the base of the palm of 3D$^{pol}$. The 1.7 Å resolution crystal structure of the FMDV 3D$^{pol}$-3B1 complex shows that 3B1 simultaneously links two 3D$^{pol}$ molecules by binding at the bottom of their palm subdomains in an almost symmetric way. The two 3B1 contact surfaces involve a combination of hydrophobic and basic residues at the N- (G5-P6, R9; Region I) and C-terminus (R16, L19-P20; Region II) of this small protein. Enzyme-Linked Immunosorbent Assays (ELISA) show that the two 3B1 binding sites play a role in 3D$^{pol}$ binding, with region II presenting the highest affinity. ELISA assays show that 3D$^{pol}$ has higher binding affinity for 3B1 than for 3B2 or 3B3. Membrane-based pull-down assays show that 3B1 region II, and to a lesser extent also region I play essential roles in mediating the interaction of 3AB with the polymerase and its recruitment to intracellular membranes.

## Author summary

The Picornaviridae family includes several well-known human pathogens such as poliovirus (PV), rhinovirus (HRV) and hepatitis A virus (HAV), and important animal

3Dpol-3B1-UTP crystals, and 8C2P for the 3DpolStop-3B3 crystals

**Funding:** This work was funded by the Spanish Ministry of Science and Innovation grant PID2020-117976GB-I00 to NV. The funders had no role in study design, data collection and analysis, decision to publish, or preparation of the manuscript.

**Competing interests:** The authors have declared that no competing interests exist.

pathogens such as foot-and-mouth disease virus (FMDV). Two key interactions in picornaviruses genome replication are those stablished between the enzyme responsible of genome replication, the RNA polymerase $3D^{Pol}$, and the viral proteins 3AB and 3B (or VPg). 3AB is a membrane-binding protein in charge of the recruitment of $3D^{Pol}$ to the membranous compartments in which replication occurs. 3B is a 22–24 amino acids soluble peptide, product of 3AB processing that serves as primer for RNA replication. Unlike other Picornaviruses, FMDV encodes three similar copies of 3B (3B1, 3B2 and 3B3) that could be specialized in different functions during replication. The structural and functional data presented in this work reveals that the preferential binding site for 3B1 is the bottom of the palm subdomain of $3D^{Pol}$ and that this binding is essential for the recruitment of $3D^{Pol}$ to membranes.

## Introduction

The *Picornaviridae* family includes a broad range of human pathogens, including polioviruses (PV), rhinoviruses (HRV), hepatitis A virus (HAV) and the emerging enteroviruses EV-A71 and EV-D68, among others [1]. The family also includes important animal pathogens such as foot-and-mouth disease virus (FMDV), the causative agent of a highly contagious disease affecting cloven-hoofed animals that has a significant global economic impact [2].

Picornaviruses are non-enveloped RNA viruses possessing a single-stranded RNA genome (7–8 kb) of positive polarity, with a small protein, named 3B or VPg (from 19 to 26 amino acids long), linked to its 5′-end. Upon infection, the genome is translated to a single polyprotein that is post-translationally cleaved by viral proteases to release the structural proteins (VP1-4), needed to assemble virus capsids and the non-structural proteins (2A-2B-2C-3A-3B-$3C^{pro}$-$3D^{Pol}$ and in some genera L) as well as some stable precursors, such as 3AB or 3CD, essential for virus replication in host cells [3]. The picornavirus genome is replicated via a negative-sense RNA intermediate by the viral RNA-dependent RNA Polymerase (RdRP) $3D^{Pol}$. The structure and function of $3D^{Pol}$ has been studied extensively in the past decades and, to date, the high resolution X-ray structures of $3D^{Pol}$ are available for several members of the family [4–14]. The enzyme adopts the canonical closed right-hand polymerase fold, consisting of fingers, palm, and thumb subdomains that encircles seven conserved structural motifs (A to G), playing critical roles in rNTP substrate recognition, template/primer binding and catalysis [15].

$3D^{Pol}$ uses 3B as a primer protein to initiate the replication process [16–17]. The very first step in picornavirus genome replication is the uridylylation of tyrosine 3 (Y3), in the 3B protein [16]. In this process, $3D^{Pol}$ catalyzes the successive attachment of two uridine monophosphate (UMP) molecules to the hydroxyl group of this tyrosine which is conserved among picornaviruses. This process has been extensively studied for different members of the *Picornaviridae* family [8,10,11,18–19]. In addition, the X-ray structures of three $3D^{Pol}$-3B complexes have been previously solved, revealing three functional 3B binding sites on $3D^{Pol}$ [10,11,20] (S1A Fig). The FMDV $3D^{Pol}$-3B1 structure showed the 3B primer bound to the central cavity of $3D^{Pol}$, projecting the conserved Y3 into the active site. This "front-loading" mode for 3B binding is compatible with a *cis* mechanism of 3B uridylylation [19] (S1A Fig). In contrast, the structure of the $3D^{Pol}$-3B complexes of CVB3 and EV71 revealed two additional binding sites for 3B: at the base of the thumb and at the bottom of the palm of their polymerases, respectively, in orientations that would only be compatible with a mechanism of 3B uridylylation in *trans* [10,11] (S1A Fig). These data indicated that 3B and $3D^{Pol}$ were able to interact in at least three different ways. However, due to the notable sequence conservation between picornaviral

3Bs and the large similarities existing among the 3D$^{Pol}$ structures, it seems reasonable to assume a similar uridylylation mechanism in all picornaviruses. Hence, the different sites identified for 3B might be related with different functions of the 3B peptide in the picornavirus replication complex.

In addition, unlike other Picornaviruses, FMDV encodes three similar copies of the 3B protein (designated 3B1, 3B2 and 3B3) and all of these were found linked to the viral genome (S1B Fig) [21,22]. The biological meaning of this surprising redundancy, considering the small viral genome size, is still under debate. Different studies have shown that the three 3Bs can be uridylylated and prime RNA synthesis [23]. Further studies have shown that multiple copies of 3B may influence host specificity and virulence [24] but a single copy is sufficient to support viral replication [25–26]. Although not all the copies are needed to maintain infectivity, there are no reports of naturally occurring FMDV strains with fewer than three copies of 3B, indicating that there is a strong selective pressure towards maintaining this redundancy [26–28]. Therefore, although the three FMDV 3Bs can perform the same tasks, it is tempting to hypothesize that each 3B may be specialized in a particular task during FMDV replication.

To shed new light on this question here we combined structural, biochemical and biophysical approaches to identify that the FMDV 3B1 peptide binds preferentially the bottom of the palm sub-domain of 3D$^{Pol}$, in a similar position to that previously described for EV71 [11]. At this position 3B1 acts as a connector between two 3D$^{Pol}$ molecules, facilitating the formation of long fibres in the crystal structure. This 3B1-3D$^{Pol}$ binding mode also facilitates the interactions of 3D$^{Pol}$ with the 3AB precursor and its recruitment to intracellular membranes.

## Materials and methods

### Protein expression and purification

FMDV 3D$^{Pol}$ proteins, with and without 6xHis tag in the C-terminus (3D$^{Pol}$His and 3D$^{Pol}$Stop, respectively), were expressed from the corresponding plasmid constructs and purified as previously described [6]. Enzymes were >95% pure, according to analytical SDS-polyacrylamide gel electrophoresis (PAGE) and Coomassie brilliant blue staining. Purified proteins were concentrated to ~5 mg/ml in a storage buffer containing, 100mM NaCl, 50mM Tris pH 8.0, 8% glycerol, 0.8mM DTT and 0.8mM EDTA.

The three 3B peptides used for crystallization: FMDV strain C-S8c1 3B1, sequence (GPYAGPLERQRPLKVRAKLPRQE), 3B2, sequence (GPYAGPMERQKPLKVKARAPVV KE) and 3B3, sequence (GPYAGPVKKPVALKVKNLIVTE) were prepared by solid-phase synthesis, purified by G25 Sephadex chromatography and HPLC, and analysed by mass spectrometry. DNA sequences coding for FMDV 3B1 [wild type and mutants, designed to confirm the two 3D$^{Pol}$ binding regions: I (P6S/ R9A), II (R16A/ L19S) and I-II (P6S/ R9A/ R16A/ L19S)], 3B2 and 3B3 were introduced by PCR into the vector pGEX-4T-2 (Cytiva) in order to produce the fusion proteins GST-3B1, GST-3B1(R16A/ L19S), GST-3B1(R16A/ L19S), GST-3B1(R16A/ L19S/ R16A/ L19S), GST-3B2 and GST-3B3. The oligonucleotides used were summarized in S1 Table. The six constructs were expressed in *E. coli* BL21(DE3) strain with 0.5mM of IPTG at 20˚C *overnight*, and purified using a GST affinity column (GSTrap FF 1ml, Cytiva) in 300mM NaCl, 50mM Tris pH7.5, 8% glycerol and eluted with 10mM of reduced glutathione. An additional purification step by size exclusion chromatography was carried out, using a superdex 75 10 300 (GE Healthcare) column. All proteins were concentrated to 10 mg/ml using a 3kDa membrane ultracentrifugal filters (Amicon) in a 100mM NaCl, 50mM Tris pH7.5 and 8% glycerol buffer. Proteins were frozen and stored at -80˚C.

## Complex formation and crystallization assays

To obtain the different FMDV 3D$^{Pol}$-3B complexes, 3D$^{Pol}$His or 3D$^{Pol}$Stop were incubated with 3B1, 3B2 and 3B3, individually or using 3B mixtures at different concentrations and, in the presence or absence of ions and UTP (S2 Table). The different solutions were incubated overnight at 4°C, previous to the crystallization trials. Crystals appeared in two or three days from solutions containing 0.2M Ammonium acetate or magnesium acetate, 25% PEG 4K, 0.1M HEPES pH 6.5 and 4% γ-butyrolactone.

Hundreds of crystals were harvested in cryo-loops (Molecular Dimensions), soaked for 1 min in a solution containing the crystallization buffer and 20% (v/v) glycerol, and flash-frozen in liquid nitrogen.

## X-ray data collection, processing and structure determination

Diffraction data were collected at 100K using synchrotron radiation on the XALOC beamline at the ALBA Synchrotron (Cerdanyola del Valles, Spain), on a Pilatus 6M DECTRIS detector. X-ray data were processed, using XDS [29] or iMosflm [30–31] and internally scaled with scala (ccp4i) [32]. Co-crystals of 3D$^{Pol}$His or 3D$^{Pol}$ Stop in complex with the different 3Bs belonged to the trigonal space group P3$_2$21 and diffracted between 1.85Å and 3.0Å resolution (S3 Table). A different crystal form, space group P2$_1$2$_1$2$_1$, was obtained from complexes 3D$^{Pol}$His-3B1- UTP, achieving 1.7Å resolution. The initial maps for the trigonal crystal structures were obtained after rigid-body fitting of the coordinates of FMDV 3D$^{Pol}$ that was crystallized in the trigonal P3$_2$21 space group (PDB id 1WNE) [6], to the new unit cells, using the program Refmac5 [33]. These maps showed the presence of extra densities at the bottom of the palm domain, where some residues of the 3B1 or 3B3 molecules were manually positioned, using the program Coot [34]. Observing the electron density maps of different complexes (3D$^{Pol}$His-3B1, 3D$^{Pol}$His-3B3 or 3D$^{Pol}$Stop-3B1, 3D$^{Pol}$Stop-3B3), we did not see differences between 3D$^{Pol}$His and 3D$^{Pol}$Stop, concluding that the His-tag did not interfere with the localization of 3B. We also observed that in the trigonal crystals both 3B1 and 3B3 were bound to the same place at the bottom of the palm, therefore the refinement was finished only with the 3D$^{Pol}$-Stop-3B3 crystals that diffracted at the highest resolution (S2 and S3 Tables). Orthorhombic P2$_1$2$_1$2$_1$ crystals contained two 3D$^{Pol}$ molecules in the asymmetric unit. The structure was solved by Molecular Replacement with the program MolRep [35], using the coordinates from the isolated FMDV 3D$^{Pol}$ (PDB id 1U09) as a search model. The initial electron density maps showed an elongated extra density at the bottom of the palm domain connecting the two polymerase molecules of asymmetric unit. The 3B1 model (amino acids from G5 to R21) was manually build in this extra density using Coot [34].

Several cycles of automatic refinement, performed with Refmac5 [33] and Phenix [36], were alternated with manual model rebuilding using Coot. The quality of the final refined models were verified using the program PROCHECK [37]. The refinement statistics and model validation parameters of the two crystal structures are given in S3 Table. Illustrations were prepared with PYMOL [38].

## Protein structure accession numbers

The coordinates and the structure factors for the complexes were deposited in the Protein Data Bank under accession numbers: 8C1N (3D$^{Pol}$-3B1-UTP) and 8C2P (3Dpol-3B3).

## Enzyme-Linked Immunosorbent Assay (ELISA)

96-well Maxisorp immunoplates (Nunc) were coated with 10 μg/ml of 3D$^{pol}$ in PBS *overnight* at 4° C. Wells were blocked with 2% BSA in PBS 1h at RT. The following steps were performed using 1%BSA PBS buffer for incubations and PBS buffer for washes. After four washes, different concentrations (from 0.5μg/ml to 20 μg/ml) of the GST, GST-3B1 (WT and mutants), GST-3B2 and GST-3B3 were added and incubated 1h at RT. Plates were again washed four times and incubated with 1.5μg/ml of anti- GST Mouse/Rabit antibody (Life Technologies), 1h RT. After four washes more, 1μg/ml anti- Mouse HRP conjugate (Life Technologies) was added at plate and incubated 1h at RT. Finally, plates were extensively washed and color reaction was developed with 100μl/well o-phenylenediamide dihydrochloride (OPD) (1mg/mL) in the presence of 0.03% $H_2O_2$ in a phosphate/citrate buffer. The reaction was stopped with 100μl/well 2M $H_2SO_4$. Absorbance was read at 492nm and corrected for the blank (sample with 3D$^{pol}$ and without GST-3B). All presented data points are means of at three duplicates in three independent experiments. Antibody non-specific binding was monitoring omitting 3D$^{pol}$, GST-3B1, anti-GST or anti-HRP and the blackground was substracted for each point from wells with the result of the wells without GST-3B1.

## Expression and purification of FMDV 3AB-containing E. coli membranes

Plasmid pET3a with the coding sequence for the precursor protein 3AB, with a FLAG tag linked at its N-terminus, was synthesized by Genscript. Mutations 3B1(P6S/ R9A), 3B1(R16A/ L19S) and 3B1(P6S/ R9A/ R16A/ L19S), were introduced by site directed mutagenesis. All mutations were confirmed by DNA sequencing. The primers used were summarized in S1 Table.

Membrane-bound FMDV 3AB WT and mutants were expressed and purified as previously described [18,39]. Briefly, *E. coli* BL21(DE3) cells (Novagen) transformed with the pET3a-3AB plasmids were grown at 37°C to an optical density at 600 nm of 0.8 in LB containing 50 mg/ liter ampicillin. The protein expression was induced with 0.5 mM IPTG and the cells were shaken at 22°C overnight. Cells were harvested by centrifugation, washed once in 50 mM Tris (pH 7.5), 100 mM NaCl, pelleted again, and stored at -20°C. Frozen pellets were thawed and resuspended in 3AB lysis buffer [5% glycerol, 50 mM Tris, pH 7.5, 100 mM NaCl, 1mM Na2-EDTA, 1 mM dithiothreitol and one tablet of cOmplete Protease Cocktail Inhibitor (Roche) for each 50ml]. Cells were lysed in a French pressure cell press at 16 Kpsi and centrifuged at 9,000 x*g* at 4°C for 40 min to remove cellular debris. The supernatant was then saved and centrifuged at 100,000 x *g* at 4°C for 30 min to collect cellular membranes. This pellet was resuspended in 3AB lysis buffer. The samples were stored at -80°C. Control membranes were purified from *E. coli* containing pGEX-4T-2 plasmid (Merck), following the same protocol. The total protein concentration was normalized for the four samples by western blot, using a mouse anti-FLAG primary antibody (sigma, M2 F3165) and a Donkey anti- Mouse IgG (H+L) secondary antibody HRP (Life technologies, A16011).

## Polymerase recruitment assay in E. coli membranes

The polymerase recruitment assay was performed based on a previous described protocol [18,39]. Briefly, reaction mixtures consisting of 10μl of 3AB-containing or control membrane (0.075 mM) in 3AB lysis buffer and 5μl of 3D$^{pol}$ (0.094 mM), in 40% of glycerol and 600 mM NaCl, were placed on ice for 60 min, incubated at 30°C for 20 min, and then spun at 14,000 rpm for 10 min in a microcentrifuge at 4°C. Pellets were then resuspended in washing buffer (25 mM Tris, pH 7.5, 500 mM NaCl, 10 mM dithiothreitol), centrifuged again as before, and finally resuspended in SDS loading buffer. Proteins were resolved by electrophoresis on a 15%

SDS-PAGE, and transferred onto a nitrocellulose membrane (Amersham) for a Western Blot. Wet electroblotting (50V, 150 mA, ON) was performed in buffer, 25 mM Tris-HCl pH 8.3, 192 mM glycine, 20% (v:v) methanol. Membranes were blocked with 5% non-fat milk in PBST (0.5% Tween 20 in PBS) during 2 hours at RT, followed by incubation with primary antibodies [anti-His tag mouse monoclonal antibody (Sino Biological) or anti-FLAG (sigma, M2 F3165) to detect 3D$^{pol}$ and 3AB, respectively], during 1 hour at RT in blocking buffer, two washes with PBS and a final incubation with secondary antibody [Donkey anti-Mouse IgG (H+L) (Life technologies)]. Western Blot quantifications were performed using FijiJ software [39].

## Polymerase recruitment assay in mammalian cells

The FLAG- 3AB1 sequence was introduced into the plasmid pEGFP-N1 (Clontech, Palo Alto, SA) between the restriction sites XhoI and NotI, removing the GFP protein. The sequence coding for 3D$^{pol}$ was cloned into the same vector between NheI and EcoRI restriction sites. In this case the GFP sequence was kept in the same reading frame as the polymerase. In both cases the KOZAK sequence was maintained to properly produce the proteins in eukaryotic cells. Mutations in 3AB1(P6S/ R9A/ R16A/ L19S) were introduced by site directed mutagenesis. All mutations were confirmed by DNA sequencing (Eurofins genomics). Primers used for cloning experiments were summarized in S1 Table (Integrated DNA Technologies).

HeLa S3 cells immortalized human cervical cancer cell line ECACC 93021013 were maintained at 37˚C/5% CO$_2$ in Dulbecco's modified Eagle's medium (DMEM) (Biowest) supplemented with 4.5 mg/ml glucose, 584 mg/l L-glutamine, sodium pyruvate (110 mg/l; Biowest, L0104-500), 1% nonessential amino acids (MERCK), 100 units/ml penicillin, 100 mg/ml streptomycin (MERCK) and 10% fetal calf serum (FCS; Biowest).

Subconfluent monolayers of HeLa cells were grown 24 hours in microscope cover glasses (12 mm diameter). These cells were transfected with the appropriate plasmid using Metafectene Pro (Biointex) as transfection reagent, according to the manufacturer's protocol. Briefly, cells were grown in a 24-well tissue culture plate in 1 ml of suitable fresh complete medium at 37˚C in a CO$_2$ incubator until growing area was 80% covered, between 18 and 24 hours approximately. Then the medium was completely removed and 0.5 ml of DMEM without FCS were added. The transfection mix (100 μl of DMEM and 2 μl of MetafectenePro were mixed) were prepared in microtubes, adding 0.5 μg of transfection plasmid. Solutions were mixed gently and incubated at room temperature for 15–20 min before adding to the cells. Transfected cells were incubated 10 hours at 37˚C in a CO$_2$ incubator and then supplement with 10% of FCS.

After 24 hours, transfected HeLa cells grown on cover glasses were fixed using a 4% paraformaldehyde solution in PBS for 30 min at room temperature. After washing the cover slips once with PBS, cells were permeabilized using 0.1% (v/v) TritonX-100 for 15 min at room temperature, followed by a 30 min incubation in blocking buffer (PBS containing 10% FCS). Subsequently, the cells were incubated for 1 hour at 37˚C with the indicated primary antibody diluted 1:1000 (Monoclonal ANTI-FLAG M2 antibody produced in mouse; F3165-.02MG; MERCK). After six washes with PBS, the cells were incubated for 1 hour with a secondary antibody marked with Alexa594 (A11032; Thermo fisher scientific), followed by six washing steps with PBS. Nucleus were stained with DAPI and the cover glasses were mounted with ProLong Gold (P10144; Life Technologies).

For the fluorescence microscopy analyses, samples were examined using a High Speed and Super -resolution Dragonfly 505 Confocal Microscope (ANDOR). Images were analyzed and quantified using ImageJ [40].

The experiments of transfection and immunofluorescence were realized repetitively and the images shown are representative of the total number of images obtained.

## Results

### Preparation and crystallization of the FMDV 3D$^{Pol}$-3B complexes

In order to investigate a possible relationship between the three different 3B binding sites in 3D$^{Pol}$ described and the three copies of 3B encoded by the FMDV genome, co-crystals of the FMDV 3D$^{Pol}$ with each individual 3B as well as with 3B1, 3B2 and 3B3 mixtures were obtained. Previous to crystallization experiments, 3D$^{Pol}$ was incubated with an excess of 3B1, 3B2, 3B3 or mixtures to ensure that all binding sites could be covered. Co-crystals were also obtained in presence of UTP and ions (S3 Table). To rule out that the 6x-His tag attached to the C-terminus of 3D$^{Pol}$, used for affinity purification, could interfere with any of the 3B binding sites, two constructs of the polymerase were produced, with and without 6xHis tag (3D$^{Pol}$His and 3D$^{Pol}$-Stop, respectively) and co-crystals were obtained using the two polymerase versions. Hundreds of crystals were analysed but only some of them diffracted at high resolution. S2 Table gives a general outline of the crystal complexes analysed, diffracting at reasonable resolution. It should be mentioned that none of the crystals obtained with 3B2 diffracted.

The best diffracting crystals were those of the 3D$^{Pol}$His-3B1 complex grown in the presence of UTP and Mn$^{2+}$. These crystals, orthorhombic P2$_1$2$_1$2$_1$, contained two 3D$^{Pol}$ molecules in the asymmetric unit and diffracted at 1.7 Å resolution (S3 Table). Different crystals were of complexes: 3D$^{Pol}$His-3B1, 3D$^{Pol}$His-3B3 or 3D$^{Pol}$Stop-3B1, 3D$^{Pol}$Stop-3B3 that belonged to the space group P3$_2$21. Several data sets were collected from these crystals at resolutions ranging from 3.0 Å to 1.85 Å (S3 Table).

The structures were solved by molecular replacement, using the coordinates of isolated FMDV 3D$^{Pol}$ (PDB id 1U09)[6] as a search model in the P2$_1$2$_1$2$_1$ crystals and the coordinates from the FMDV 3D$^{Pol}$ -RNA complex (P3$_2$21 crystals; PDB id 1WNE)[6], as a search model in the trigonal crystals (S3 Table). In both crystals, the initial electron density maps showed the presence of elongated extra densities at the base of the palm subdomains of 3D$^{Pol}$, where the 3B peptide could be positioned (Figs 1A and 1B and S2). These results indicated that the base of the palm was also a binding site for the 3B peptide to the FMDV 3D$^{Pol}$. The quality of the electron density for the 3B1 molecule in the P2$_1$2$_1$2$_1$ crystals allowed to interpret with confidence the 3B1 amino acids from G5 to R21 (Fig 1B). While in orthorhombic crystals the bound 3B1 peptide connected the two 3D$^{Pol}$ molecules of the asymmetric unit (Fig 1A), in the P3$_2$21 space group, the proximity of a neighbour 3D$^{Pol}$ molecule in the crystal packing (S3 Fig) prevented the correct binding of the whole 3B peptide, and only a few residues could be positioned in the electron density maps (S2A Fig). In these crystals, the position of the visible 3B1 fragment is almost coincident with that of 3B3 (from G5 to P10), therefore, the final structural analysis was carried out only with the 3D$^{Pol}$Stop-3B3 complex structure that diffracted at the highest resolution (1.85 Å; S3 Table).

### 3B1 binds the base of the FMDV 3D$^{Pol}$ palm, acting as a glue between two polymerase molecules in the P2$_1$2$_1$2$_1$ crystals

In the structure of the 3D$^{Pol}$His-3B1 complex, 3B1 molecule binds the bottom of the palms of two confronted 3D$^{Pol}$ molecules (named 3D$^{Pol}$I and 3D$^{Pol}$II), in an almost equivalent way (Fig 1). The 3B1 binding region in both molecules consists in a hydrophobic cavity formed by residues: from H322 to V326, T330 and from Y346 to L348. These cavities are occupied by two hydrophobic residues (G5 and P6; region I) and (L19 and P20, region II) located at the N- and

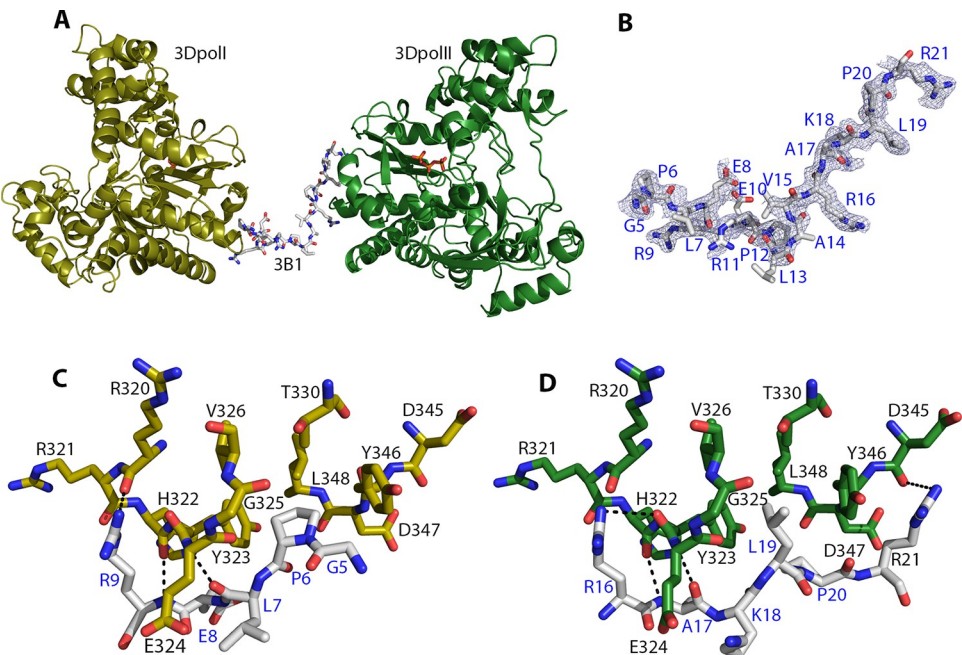

**Fig 1. Structure of the FMDV 3D^Pol-3B1 complex.** (A) Ribbon representation of the two 3D^Pol molecules present in the asymmetric unit of the P2₁2₁2₁ crystals (yellow and green ribbons for the 3D^Pol I and 3D^Pol II molecules, respectively). The 3B1 molecule bound to the bottom of the palm of the two 3D^Pol molecules is shown as sticks in atom-type colour (carbons white). The ordered triphosphate moieties of the UTP molecules bound at the nucleotide-binding site of the polymerases are also shown in atom-type sticks (phosphates in orange). (B) Omit map around the 3B1 molecule, displayed at a contour of 1,0 σ (light blue mesh). 3B1 is shown in sticks as in A with the amino acids explicitly labelled. (C) Details of the intermolecular contacts stablished between 3D^Pol I and the 3B1 N-terminus (Interface I). (D) Intermolecular interactions between 3D^Pol II and the 3B1 C-terminus (Interface II). The interacting residues are shown in sticks representation, coloured as in A. Hydrogen bonds and salt bridges are shown as dashed lines in black.

C-terminus of 3B1, respectively (Fig 1C and 1D). In both regions, the central hydrophobic interface is surrounded by polar interactions. In region I these interactions involve the main and side chains of the 3B1 residue R9 and the main chain oxygens of residues R320 and E324 in 3D^PolI. The contact interface between 3D^Pol and 3B1 in region I, as calculated with PISA [41], was 362.9Å². In region II, an equivalent interaction is established between the 3D^Pol II main chain residues H322 and E324 and the main and side chains of R16 in 3B1. Region II also shows an additional polar contact involving the side chain of residue R21 in 3B1 and the main chain carboxyl oxygen of D345 in 3D^Pol (Fig 1C and 1D), slightly increasing the interaction surface (508.2Å) at this binding site.

Furthermore, the two 3D^Pol molecules linked by 3B1, interact with each other through contacts mediated by residues from G125 to P141 in the fingers sub-domain, forming long fibers along a diagonal in the ab plane in the crystal packing (Fig 2). The P141 residues from the two quasi-symmetric 3D^Pol molecules contacted face to face by hydrophobic interactions. In addition, the side chain amino-groups of K126 and R127 from 3D^PolI appeared salt-bridged with the side chain carboxylic groups of the 3D^PolII residues D133 and E135, respectively (Fig 2B). Beside the interactions formed between the 3D^Pol dimers each individual 3D^Pol molecule contacts six additional neighboring molecules, stabilizing the crystal packing (S4 Fig).

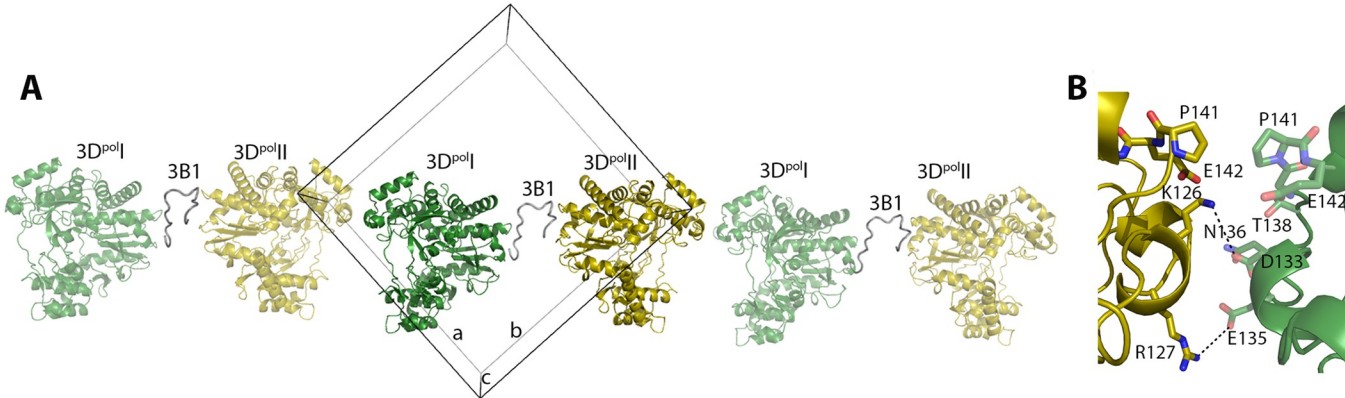

**Fig 2. The FMDV 3D$^{pol}$-3B1 complex forms long fibres in the crystal packing.** The two 3D$^{pol}$ molecules (green and yellow cartoons) linked by 3B1 (grey ribbon) also interact with each other in the crystal packing through contacts between the finger subdomains, forming long fibres along the a-b diagonal. (B) Close up of the interactions involving the direct 3D$^{pol}$-3D$^{Pol}$ contacts that facilitate fibre formation.

## The 3B-binding cavity at the base of the palm of 3D$^{pol}$ is conserved among picornaviruses

As expected, the visible part of the FMDV 3B3 peptide in the trigonal crystals bound to the same 3D$^{pol}$ cavity at the base palm, close to residues from Y323 to V326 and from Y346 to L348 (S2A Fig). This binding cavity was also equivalent to the so-called "contact II region" in enterovirus 71 (EV71) 3D$^{pol}$, involving residues: from T313 to I317 and from Y335 to P338 that anchor the second half of the EV71 3B peptide, amino acids from L11 to R17, (Figs 3A and 3C and S1B) [11].

The notable structural conservation between these 3B binding sites between EV71 and FMDV prompted us to extend the structural comparisons to other picornaviruses whose structure is known (Fig 3). These comparisons show that despite the low amino acid sequence conservation, all 3D$^{pol}$ structures show similar hydrophobic pockets where the 3B peptide could be anchored, suggesting that the base of the 3D$^{pol}$ palm is a conserved 3B binding site which may play an important role in picornavirus replication. An additional inspection of the 3B sequences from different Picornaviruses (S5 Fig) show that, with the only exception of the 3B3 from FMDV, all of them exhibit two hydrophobic clusters formed by P and V or L, each one located at the N- and at the C-terminal ends of the 3B peptides, respectively (S5 Fig).

## FMDV 3B1 residues P6 and R9 in interface I, and R16 and L19 in interface II are essential for 3B1 binding at the base of the palm of FMDV 3D$^{pol}$

To confirm the role of the two 3B1 contact interfaces in 3D$^{pol}$ binding we generated three different mutants with changes at the interacting regions: 3B1(P6S/R9A), 3B1 (R16A/L19S) and 3B1(P6S/R9A/R16A/L19S) and tested their effect on 3D$^{pol}$ binding by ELISA assays. We had previously described the binding of 3B1 (uridylylated and a non- uridylylated forms) in the central cavity of 3D$^{pol}$, approaching the essential Y3 side chain to the 3D$^{pol}$ catalytic site for uridylylation [20]. It has also been reported that a free N- terminus is required for 3B function as a primer [27]. Taking these works into account and to prevent the binding of 3B1 to the 3D$^{pol}$ central cavity, 3B1 was cloned, expressed and purified 3B1 bound to GST at its N- terminus. GST-3B1 wild type and mutants GST-3B1(P6S/R9A) (that abolish the interaction of the region I), GST-3B1(R16A/L19S) (that inhibit the region II interface) and GST-3B1(P6S/R9A/R16A/L19S) were produced, purified and tested for their ability to bind the base of the palm of

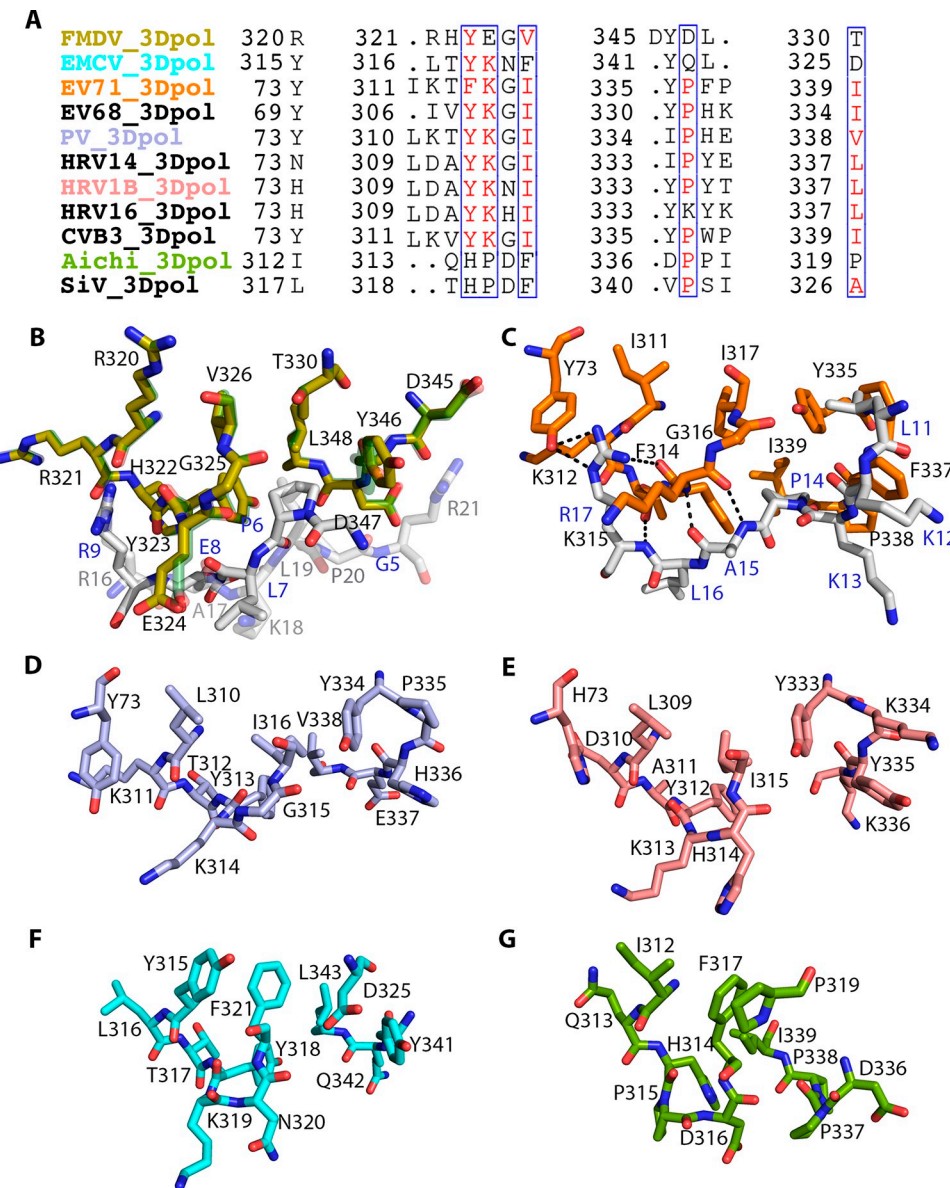

**Fig 3. Conservation of the 3B contact surface at the base of the palm of 3D^pol among picornaviruses.** (A) Structure-based sequence alignment of the picornavirus 3D^pol residues located the base of the palm that would participate in interactions with 3B, (B) Structural superimposition of the two quasi-equivalent 3B1 binding sites in FMDV 3D^pol. The polymerase residues and the bound 3B1 regions are shown in sticks, coloured as in Fig 1, but with molecule I shown in semi-transparent. (C) The 3B binding site in EV71 3D^pol, as seen in the X-ray structure of the EV71 3D^pol -3B complex [11] (PDB:4IKA). (D-G) Structural comparisons of the putative 3B binding region in 3D^pol of other representative picornaviruses whose structure is known: the enteroviruses PV [5] (PDB: 1RA7; light blue) **(D)** and HRV1B [7] (PDB: 1XR6; salmon) (**E**), the cardiovirus EMCV [14] (PDB: 4NYZ; cyan) (F), and the kobusvirus porcine aichi virus [13](PDB: 6R1I; green).

3D^pol in an ELISA assay (Fig 4A). The results clearly show that even under increasing concentrations of 3B1 (R16A L19S), the binding to immobilized 3D^pol barely reached the ~30% in comparison to WT 3B1 tested under identical conditions. This is strong evidence that 3B1 region II contain the main 3D^pol binding site, because mutations in 3B1 region I only reduce

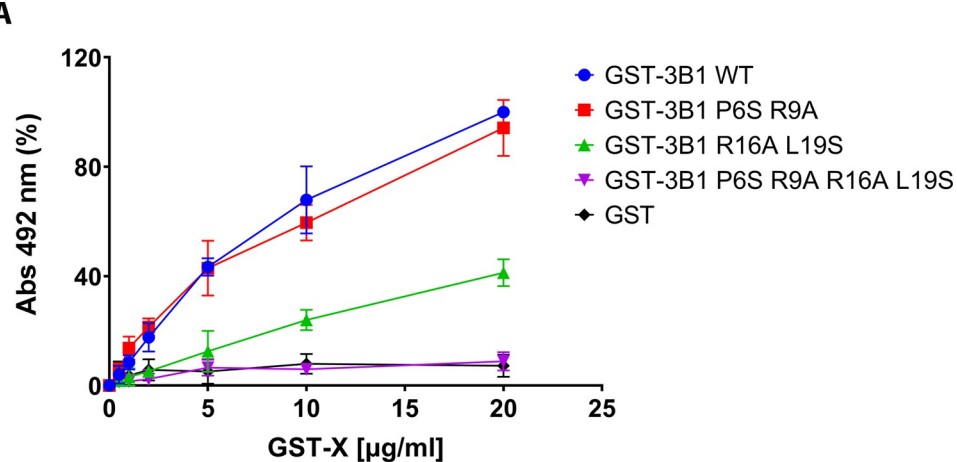

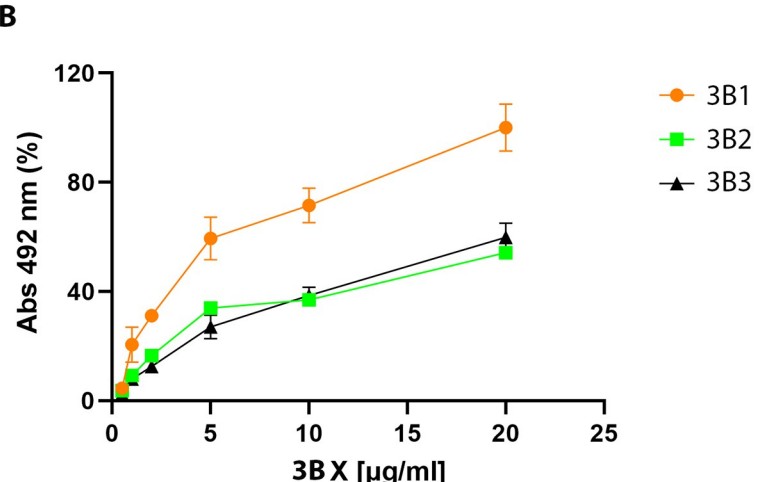

**Fig 4. 3B1 binds the bottom of the 3D$^{pol}$ palm also in solution.** (A) Enzyme-Linked Immunosorbent Assay (ELISA) to compare the binding affinities of 3D$^{pol}$ I and 3D$^{pol}$ II binding sites of 3B1, determined in the X-ray structure. A multi-well plate was coated with 10 μg/ml of 3D$^{pol}$ and its interaction to GST-3B1 WT was measured at increasing concentrations (from 0.5μg/ml to 20 μg/ml) and compared with mutants: GST-3B1(P6S/R9A), GST-3B1(R16A/L19S), GST-3B1(P6S/R9A/R16A/L19S) and GST. (B) ELISA assay to compare the affinities of 3B1, 3B2 and 3B3 for 3D$^{pol}$. Data were obtained from three independent experiments and standard deviations are reported.

10% the interaction at highest protein concentration. As expected, the cumulative 3B1 mutant was completely unable to bind 3D$^{pol}$ under this ELISA conditions (Fig 4A).

## GST-3B1 binds preferentially the base of the palm of FMDV 3D$^{pol}$ compared to GST-3B2 and GST-3B3

To maintain gene duplication should offer some advantages, especially in the small genomes of RNA viruses. Previous *in vitro* assays demonstrated that FMDV 3D$^{pol}$ uridylylates preferentially synthetic 3B3 over 3B1 and 3B2 [23,27]. Assuming that the access to the 3D$^{pol}$ active site binding site is blocked by GST fusion, here we determined which of the three copies of FMDV 3B has preference for binding at base of the palm of 3D$^{pol}$, using ELISA assays. As shown in

Fig 4B the binding affinity of 3B1 for 3D$^{pol}$ is twice that of 3B2 or 3B3. This result agrees well with the structural (Fig 1C and 1D) and sequence data (S2 Fig). As seen in the previous sections the 3B1 residue L19 plays a crucial role in 3D$^{pol}$ binding (Fig 1D). This amino acid is replaced by A in 3B2 (S5 Fig), reducing the hydrophobic contacts in the 3D$^{pol}$ binding cavity. Also, L19 is replaced by N in 3B3 (S5 Fig), hindering the placement of this residue in the hydrophobic pocket. In light of these data we suggest that 3B2 and 3B3 could bind 3D$^{pol}$ through region I which has a lower affinity (Figs 4B and S1A).

## 3B1 is essential for the recruitment of 3D$^{pol}$ to E. coli membranes

Data obtained in previous sections showed a specific binding site of 3B1 at the base of the palm of 3D$^{pol}$, indicating a critical role of this interaction in genome replication. To validate the importance of the 3B1-3D$^{pol}$ contact interface for the recruitment of FMDV 3D$^{pol}$ to intracellular membranes, polymerase recruitment assays were performed, using *E. coli* membranes that contain the FMDV 3AB1 precursor, following a protocol previously described for PV [18,39]. Analyses of 3D$^{pol}$-binding by wild type 3AB1 and mutants disrupting the 3B1-3D$^{pol}$ contact surfaces indicate that 3AB1(R16A L19S), affecting the contact surface II, has lost the ability to bind 3D$^{pol}$ in more than 95% and no binding was observed in the mutant 3AB1(P6S R9A R16A L19S) (Fig 5). Finally, mutant affecting the contact surface I, 3AB1(P6S R9A), displays a less dramatic effect, reducing the 3AB1-3D$^{pol}$ binding levels to about 30% compared to the wild-type (Fig 5).

## 3D$^{pol}$ is recruited to 3AB1-bound membranes in Hela cells

To better characterize the role of 3B1-3D$^{pol}$ interaction for polymerase recruitment to intracellular membranes, the subcellular distribution of both proteins and mutants were studied in

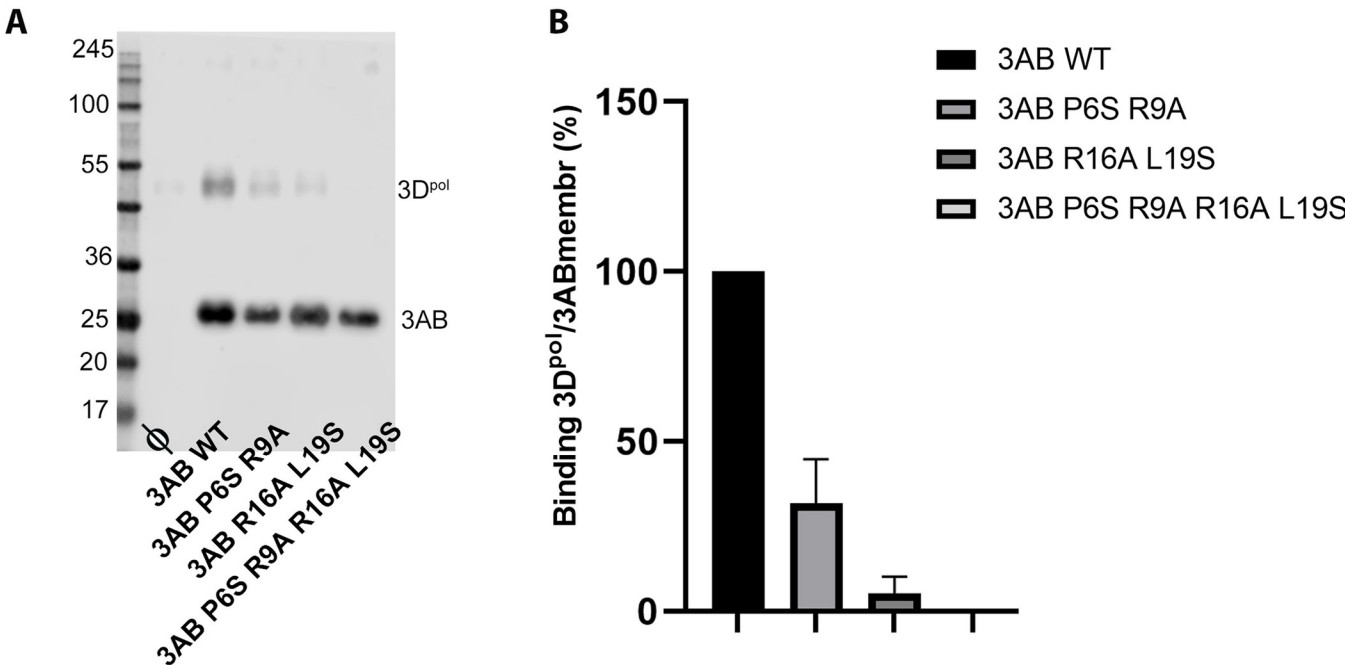

**Fig 5. Polymerase recruitment to *E. coli* membranes containing the 3AB1 precursor.** *E.coli* membranes containing, wild-type 3AB1 or 3AB1(P6S/R9A), 3AB1(R16A/L19S) or 3AB1(P6S/R9A/R16A/L19S) mutants, disrupting the 3B1-3D$^{pol}$ binding site were used in the analysis. (A) Western blot experiment that shows the 3D$^{pol}$ protein recruited to the membrane. (B) Bars diagram indicating the percentage of 3D$^{pol}$ bound by the membranes. To calculate the fraction of 3D$^{pol}$ bound to 3AB, the volume of the 3D$^{pol}$ band from a control membrane was subtracted from each of the other 3D$^{pol}$ bands, and these were then normalized to the 3D$^{pol}$ band pulled down by wild-type 3AB. Assays were performed in triplicate, and the mean value and the standard deviation are reported.

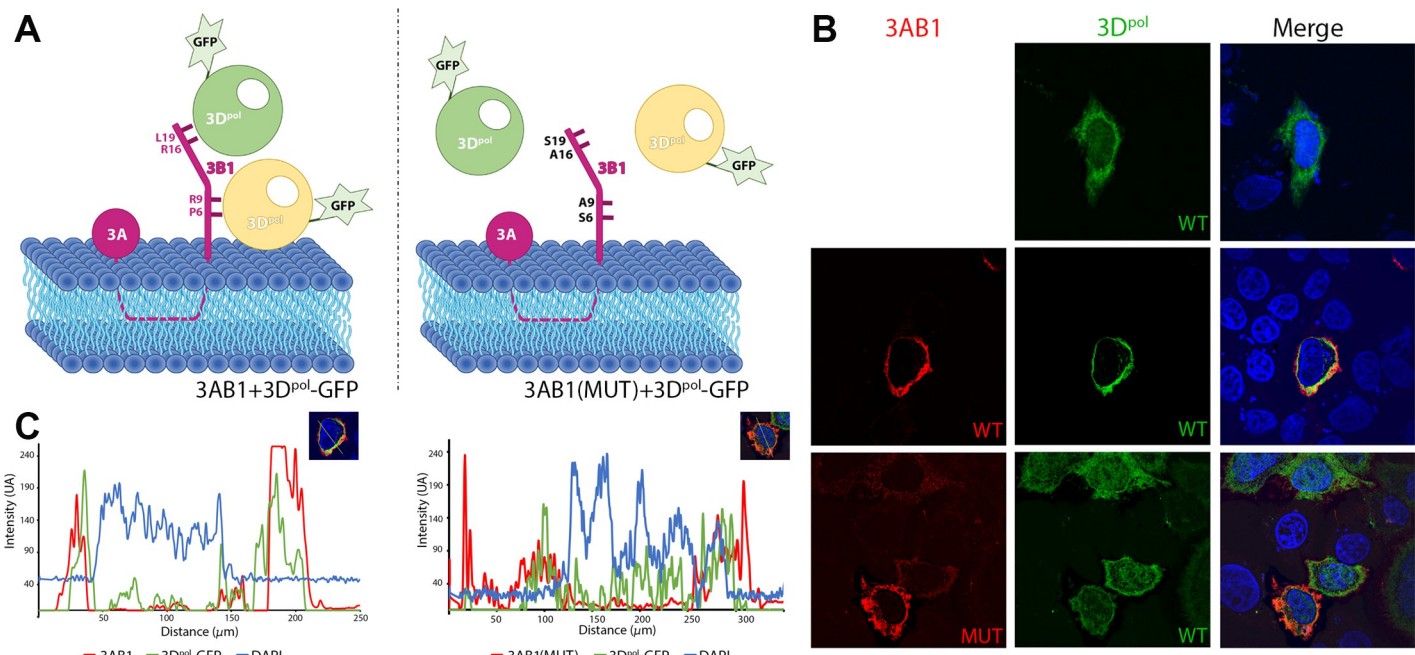

**Fig 6. The binding of 3AB1 at the bottom of the palm subdomain of 3D^pol increases localization of polymerase to intracellular membranes.** (A) Scheme representing the fluorescence microscopy experiments comparing the distribution of 3D^pol in the presence of 3AB1, wild type and mutant. The left panel shows 3D^pol interacting with the 3AB1 precursor bound to the membrane. The right panel mimics the scenario in the presence of the 3AB1 mutant 3AB1(P6S/R9A/R16A/L19S), unable to bind 3D^pol. (B) Fluorescence images of HeLa cells showing the different distribution of 3D^pol bound to 3AB1 wild type or the 3AB1(P6A/R9A/R16A/L19S) mutant. Upper panels show the control cells transfected with the polymerase only (green), which appears distributed throughout the cell. Middle panels show cells transfected with 3AB1 wild type (red) and 3D^pol (green), where 3D^pol mostly co-localizes with the 3AB1 protein in a continuous compartment in the cytoplasm. The lower panels show cells transfected with the 3D^pol and the 3AB1 mutant (P6A/R9A/R16A/L19S), where 3D^pol recovers its localization throughout the cell. The images shown are representative of the total number of images obtained. (C) Fluorescence Intensity plots comparing the relative distribution of 3D^pol (green) in presence of wild type and mutant 3AB1 proteins (red), left and right panels, respectively. The nucleus is labelled in blue (DAPI).

HeLa cells by transient expression and IF labeling (Fig 6A). The single EGFP-3D^pol expression showed a diffuse protein distribution that include cytoplasm and nuclear compartment (Fig 6B, upper panels). No evident nuclear morphology alterations were observed. However, the co-expression of flag tagged 3AB1 protein precursor showed a defined perinuclear distribution (Fig 6B, middle panel left) compatible with its reported localization as a transmembrane protein in organelle-derive mixtures [42]. Protein co-expression also revealed a drastic change in EGFP-3D^pol pattern distribution. 3D^pol mostly co-localize with 3AB1 in defined cytoplasmic regions (Pearson correlation coefficient, PCC = 0.65) and was excluded from nucleus (Fig 6B middle panels), showing that 3AB1 was recruiting the co-expressed 3D^pol to membrane localization. By contrast, 3D^pol distribution was not altered when 3AB1 (P6A/R9A/R16A/L19S) mutant was co-expressed. 3D^pol was more diffusely distributed (PCC = 0.26) even in the nucleus resembling its pattern in absence of 3AB1 (Fig 6B, lower panels). In addition, point mutations in 3B1 did not change the localization of 3AB precursor in comparison to the WT protein.

## Discussion

All positive-strand RNA (+RNA) viruses, including picornaviruses, remodel intracellular membranes of host cells to form viral replication compartments that support replication of the viral genomes [43]. These organelles protect the viral components from the host immune response and serve as a platform on which proteins involved in genome replication are

concentrated and assembled into active replication complexes. It has been described that in enteroviruses, the polypeptide responsible for binding and recruiting the RdRP 3D$^{pol}$ to the surface of these membranous compartments is the protein precursor 3AB (20kDa). 3AB binds to the replication compartments through a 22-residue hydrophobic region which inserts into the membrane, leaving both the N- (the soluble moiety of 3A) and C- (the 3B peptide) termini of the protein on the cytoplasmic side of the compartment. Further proteolytic processing of 3AB yields 3A, which remains membrane bound by its hydrophobic region, and the soluble peptide primer 3B (or VPg) [39,44–45]. In a previous work we determined the structure of two complexes between FMDV 3D$^{pol}$ and the uridylylated and non-uridylylated forms of 3B1. The structures showed the primer protein bound to the RNA binding central cleft of 3D$^{pol}$ positioning the hydroxyl group of Y3 as a molecular mimic of the free 3′-hydroxyl group of a nucleic acid primer at the active site for uridylylation [20]. This is achieved by multiple interactions between 3B1 and a number of residues located in motif F and helix α8 of the 3D$^{pol}$ fingers, and helix α13 of the thumb domain. The role of the interacting surfaces was further confirmed by functional assays with 3D$^{pol}$ and 3B mutants harboring substitutions in the amino acids involved in contacts that showed important effects in uridylylation [20].

In this work we performed the structural and functional characterization of a new biologically relevant binding site for the FMDV 3B1 protein at the base of the palm subdomain of the polymerase. This binding site involves the 3D$^{pol}$ residues from Y323 to V326 and from Y346 to L348 that form a hydrophobic cavity where 3B1 is anchored. Two 3D$^{pol}$ molecules were bound to the 3B1 peptide, stablishing quasiequivalent interactions with two 3B1 regions: I (residues from G5 to R9) and II (residues from R16 to R21) (Fig 1C and 1D). The two contact surfaces combined both hydrophobic and polar interactions. In region I, the hydrophobic pair G5, P6 connected the polymerase residues Y323, G325, V326 and L348, and the polar interactions involved the main and side chain of residue R9 in 3B1 and the main chain oxygens of residues R320 and E324 in 3D$^{pol}$ (Fig 1C). In region II, residues L19 and P20 were bound the same region of the second 3D$^{pol}$ molecule (amino acids Y323, G325, V326 and L348), and this central hydrophobic cluster was surrounded by the intermolecular polar bonds between the main and side chains of R16 in 3B1 and the main chain residues H322, E324 in 3D$^{pol}$, and the R21 side chain in 3B1 and the main chain carboxyl oxygen of D345 in 3D$^{pol}$ (Fig 1D). All FMDV 3D$^{pol}$ residues participating in the two 3D$^{pol}$-3B1 contact interfaces were equivalent to those involved in the so-called "interacting region II", described in the crystallographic structure of EV71 3D$^{pol}$-3B complex [11] (Figs 3A and 3B and S2B). In addition, structural comparisons revealed that the 3B binding cavity at the base of the palm of 3D$^{pol}$ is a conserved feature among picornaviruses (Fig 3B–3G).

To further explore the importance of the two 3B1 binding sites seen in the crystal structure, we generated mutants of the main interacting residues and tested their effect on 3D$^{pol}$ binding in ELISA assays (Fig 4). The results show that the region I mutant (GST-3B1P6S/R9A) did not produce a significant effect while the region II mutant (GST-3B1R16A/L19S) drastically reduced the binding, indicating that region II was crucial for the 3D$^{pol}$ interaction (Fig 4).

Previous *in vitro* experiments in PV showed that despite 3AB included all residues of the 3B peptide, this precursor could not be uridylylated by 3D$^{pol}$ [39]. These studies also showed that while the hydrophobic region of 3A would be the responsible for membrane binding, the 3B moiety would be required for binding to the RdRP 3D$^{pol}$ [39]. In this way, the precursor 3AB would act as a bridge between the membrane and the polymerase. These authors, by combining mutational analyses and membrane-based pull down assays, demonstrated that the PV 3B residues P14 and R17 were essential for the binding of 3AB to 3D$^{pol}$ and for the recruitment of 3D$^{pol}$ to membranes [39]. Equivalent experiments performed in this work with the FMDV proteins also showed that 3B1 was essential for the recruitment of 3D$^{pol}$ to membranes (Fig 5),

and that mutations at the 3B1 residues R16 and L19 (contact surface II) and to a lesser extent P6 and R9 (contact surface I) drastically disrupt the 3D$^{pol}$ recruitment (Fig 5). These results were further validated by examining the subcellular localization of 3D$^{pol}$ in Hela cells in the presence of 3AB1, wild type and the mutant 3AB1(P6A/R9A/R16A/L19S) (Fig 6). The FMDV 3D$^{pol}$-3B1 interactions in region II are essentially coincident with those described in the also called region II in the EV71 3D$^{pol}$-3B complex [11], where the contacts mediated by the 3B amino acids P14, R17 seem to be essential [11] (Fig 3C). This 3B region as well as the hydrophobic cavity at the base of the palm subdomain of 3D$^{pol}$ appeared reasonably conserved among picornaviruses (Figs 3B–3G and S5). Altogether this information indicates that the 3B binding site located at the bottom of 3D$^{pol}$ palm would be the binding site of the 3AB precursor at that the role of this binding is the recruitment of 3D$^{pol}$ to intracellular membranes (Figs 5 and 6) for replication to take place.

As mentioned in previous sections, FMDV unlike other picornaviruses, contains three similar but not identical copies of 3B in its genome [24,26]. Although viruses with a single copy of 3B can support genome replication [25–26], there is a strong selective pressure to maintain all 3B copies. In light of the existing information and the results obtained in this work, we could propose that even though the three 3B proteins are able to carry out the same functions, each individual 3B would be specialized in a specific function or would carry out this function with greater efficiency. In this way, a virus containing the three 3Bs would be better adapted. As previously described, 3B3 will be uridylylated more efficiently than the other two 3Bs [27], and here we demonstrated that 3B1 binds more efficiently to the palm base of 3D$^{pol}$ than 3B2 and 3B3 (Fig 4) and that this binding is essential for the recruitment of 3D$^{pol}$ to membranes.

Furthermore, the bottom of the palm subdomain also appears to be involved in the formation of functional oligomeric arrays of the polymerases in different picornaviruses [46] and also in caliciviruses [47]. Analysis of packing interactions in the P2$_1$2$_1$2$_1$ crystals reported here also show that the two 3D$^{pol}$ molecules bound to 3B1 interacts with other 3D$^{pol}$ dimers through direct 3D$^{pol}$-3D$^{pol}$ contacts, involving different residues of the fingers subdomain (Fig 2). These interactions result in the formation of long fibers along the crystal (Fig 2). The formation of 3D$^{pol}$ fibers during FMDV replication had also been previously described [48,49]. A number of amino acids located at the base of the 3D$^{pol}$ palm, found in contact with 3B1, in particular E324, Y346 and D347 (Fig 3B), appeared also involved in fiber formation [49]. In light of all these data, it would be reasonable to hypothesize that, in addition to recruiting 3D$^{pol}$ to intracellular membranes, 3B1 also participates in the formation of oligomeric arrays of the polymerase, contributing at the formation of the replication organelles bound to the membrane that concentrate the essential elements involved in replication, increasing the RNA polymerization efficiency.

Unfortunately, the preferential 3D$^{pol}$ binding site for 3B2 and the role of this binding in virus replication remains to be elucidated. The structure of the CVB3 3D$^{pol}$ in complex with 3B provided evidences for the existence of a third binding site for 3B on the back side of 3D$^{pol}$. Only the central part of 3B was visible, residues 7–15, bound to a cavity at the base of the thumb subdomain of 3D$^{pol}$ [10]. Despite the efforts made in this work (S2 Table), we did not obtain diffracting crystals of the FDMV 3D$^{pol}$-3B2 complex. Hence, we cannot confirm or rule out that the base of the thumb subdomain of FMDV 3D$^{pol}$ would be the binding site of this peptide. The transient or highly dynamic nature of the 3D$^{pol}$-3B2 interaction could be the reason why we have not been able to capture these interactions in the crystallization experiments.

## Supporting information

**S1 Fig. Comparison of three 3B binding sites previously reported in picornavirus 3D$^{pol}$.** (A) FMDV 3B-3D$^{pol}$ complex showing the primer peptide in green bound to active site cleft of

the polymerase in yellow [20](PDB id. 2F8E), the CVB3 3B-3D$^{Pol}$ complex, showing 3B bound to the back side of the polymerase (in slate) [10] (PDB id. 3CDW) and the EV71 3B-3D$^{Pol}$ complex bound to the base of the polymerase palm (in sand) [11] (PDB id. IKA4). (B) Schematic drawing of the FMDV genome.
(TIF)

**S2 Fig. The 3D$^{Pol}$-3B complexes.** (A) Cartoon representation of the structure of the FMDV 3D$^{Pol}$-3B3 complex from trigonal crystals, space group P3$_2$21. 3D$^{Pol}$ is shown in ribbons (yellow) and the VPg3 fragment, bound at the bottom of the palm subdomain, is shown as atom type stick (carbons in white). The corresponding 2Fo-Fc electron density map (1.0σ) is also shown around the 3B3 molecule as a light blue mesh. The proximity of a neighbouring 3D$^{Pol}$ molecule (Grey) in the crystal packing probably limits the 3B molecule from being arranged in an ordered way. (B) The Structure of the EV71-3B in complex with 3D$^{Pol}$. 3D$^{Pol}$ is shown in orange ribbons and the bound VPg is depicted as atom-type sticks at the bottom of palm subdomain [11] (PDB id 4IKA). (C) Detail of the interactions in the FMDV 3D$^{Pol}$-3B3 complex.
(TIF)

**S3 Fig. Packing of the FMDV 3D$^{Pol}$-3B3 complex in the P3$_2$21 crystals.** (A) 3D$^{Pol}$- 3D$^{Pol}$ interactions in the AB plane. The reference molecule is shown in green cartoons and the contacting neighbours in grey. (B) Table showing the different contact surfaces calculated with the PISA software [41]. (C) Close up views showing the main interacting regions
(TIF)

**S4 Fig. Packing of the FMDV 3D$^{Pol}$-3B1 complex in the P2$_1$2$_1$2$_1$ crystals.** (A) Table showing the different contact surfaces calculated with PISA [41]. (B) Two different of the views of the packing contacts (related by a 90˚ rotation). The unit cell represented as a reference. The long 3D$^{Pol}$-3B1 fibers, formed along the ab diagonal, are highlighted in green and yellow as in Fig 2, and the contacting neighbours in grey. (C) Close up views showing the main interacting regions.
(TIF)

**S5 Fig. Sequence alignment of the 3B proteins of the different picornaviruses.** The strictly conserved residues are in red blocks and similar residues in red characters. The FMDV 3B1 residues interacting with FMDV 3D$^{Pol}$ are marked by green asterisks. Residues of EV71 3B previously shown to contact the bottom of the palm of EV71 3D$^{Pol}$ in the X-ray structure of the complex [11](PDB:4IKA) are highlighted in yellow boxes.
(TIF)

**S1 Table. Summary of the oligonucleotides used.**
(DOCX)

**S2 Table. FMDV 3D$^{Pol}$-3B complexes.** Summary of FMDV 3D$^{Pol}$-3B complexes prepared and the number of crystals analysed that allowed obtaining X-ray diffraction data of sufficient quality to solve the structures.
(DOCX)

**S3 Table. Data collection and refinement statistics.** † Rwork = Σhkl ||Fobs(hkl)|—|Fcalc(hkl)|| / Σhkl |Fobs(hkl)|, where Fobs and Fcalc are the structure factors, deduced from measured intensities and calculated from the model, respectively. ‡ Rfree = as for Rwork but for 5% of the total reflections chosen at random and omitted from refinement.
(DOCX)

## Acknowledgments

The crystallization experiments were done at the Platform for Automated Crystallization (PAC joint IBMB-CSIC/IRB-Barcelona) with the collaboration of PAC staff. X-ray data were collected at ALBA, beamline Xaloc (Cerdanyola del Vallès, Spain) with the collaboration of ALBA staff

## Author Contributions

**Conceptualization:** Cristina Ferrer-Orta, Nuria Verdaguer.

**Formal analysis:** Cristina Ferrer-Orta, Diego S. Ferrero, Nuria Verdaguer.

**Funding acquisition:** Nuria Verdaguer.

**Investigation:** Cristina Ferrer-Orta, Diego S. Ferrero.

**Supervision:** Nuria Verdaguer.

**Validation:** Cristina Ferrer-Orta.

**Visualization:** Cristina Ferrer-Orta.

**Writing – original draft:** Cristina Ferrer-Orta, Nuria Verdaguer.

**Writing – review & editing:** Cristina Ferrer-Orta, Diego S. Ferrero, Nuria Verdaguer.

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
