## [Decision Letter · Decision Letter 0]

10 Mar 2023

Dear Prof. Verdaguer,

Thank you very much for submitting your manuscript "Dual role of the foot-and-mouth disease virus 3B1 protein in the replication complex: as protein primer and essential to recruit 3Dpol to membranes" for consideration at PLOS Pathogens. As with all papers reviewed by the journal, your manuscript was reviewed by members of the editorial board and by several independent reviewers. The reviewers appreciated the attention to an important topic. Based on the reviews, we are likely to accept this manuscript for publication, providing that you modify the manuscript according to the review recommendations.

Please in particular address the requests of the reviewer #2 for additional controls and experiments strengthening the validity of the reconstruction of 3AB-3D complexes on bacterial membranes (Fig. 5)

Sincerely,

George A. Belov, PhD

Academic Editor

PLOS Pathogens

Ana Fernandez-Sesma

Section Editor

PLOS Pathogens

Kasturi Haldar

Editor-in-Chief

PLOS Pathogens

orcid.org/0000-0001-5065-158X

Michael Malim

Editor-in-Chief

PLOS Pathogens

orcid.org/0000-0002-7699-2064

Reviewer Comments (if any, and for reference):

Reviewer's Responses to Questions

**Part I - Summary**

Reviewer #1: The manuscript by Ferrer-Orta et al provides insight into interactions between the RNA-dependent RNA polymerase (3Dpol) and primer peptide (3B1) of FMDV. Data are clearly presented and well explained.

Reviewer #2: The manuscript by Ferrer-Orta et al., describes studies on the interaction between the RNA-dependent RNA polymerase (3Dpol) of foot-and-mouth disease virus (FMDV) and different forms of the 3B peptides; FMDV is unique in expressing 3 different forms of this peptide that can each be uridylylated and used as a primer for RNA synthesis. The studies conducted here identify binding sites on the 3Dpol for the 3B peptides and suggest two different roles of the peptide, anchoring the 3Dpol to membranes as well as acting as primer. It seems most likely that membrane anchoring is also facilitated by the precursor 3AB1 since the 3A protein has a long hydrophobic region. Although some studies are performed using 3AB within bacterial cell membranes (Fig 5), some of the other studies (e.g. binding assays in Fig. 4) could have usefully included the 3AB protein as well as the 3B peptides. The assembly of the complex to achieve uridylylation of 3B also requires the 3CD precursor and the role of this protein in the interactions is not studied unfortunately.

Reviewer #3: Picornaviruses rely on uridylyation of the viral 3B protein to generate a protein primer for genome replication in a reaction that is catalyzed by the viral 3Dpol RNA-dependent RNA polymerase. There are three structures containing partial 3B peptides bound to three different polymerases, but these are at three different locations on 3Dpol and there is no clear consensus about how this critical step of picornaviral replication is carried out. Thus, despite 20+ years of 3Dpol structural biology work, the molecular details of how the 3B protein interacts with the polymerase are largely a mystery at both structural and mechanistic levels.

This manuscript by Ferrer-Orta et al. provides important new structural and biochemical information about the 3Dpol-3B interaction using the foot-and-mouth-disease virus system that is also fascinating in that it has three different tandem copies of the 3B protein. The paper presents two new crystal structures showing how 3B interacts with a pocket at the base of the polymerase palm domain. They further validate this structural observation with ELISA based in vitro binding data and cell microscopy imagining studies showing that a normally diffusely spread cytoplasmic 3Dpol polymerase is colocalized to membranes when co-expressed with a membrane anchored viral 3AB protein. In addition, mutation of key residues forming structural interactions also eliminate the binding signal from the ELISA and cell imaging studies, providing a important control that these new structures are presenting a biologically valid interaction.

**Part II – Major Issues: Key Experiments Required for Acceptance**

Reviewer #1: none

Reviewer #2: 1) For the ELISA (Figure 4A), it would have been good to include GST alone, as a negative control as well as GST-3AB, if possible (as indicated above). The text, on lines 409 and 416, refers to binding affinity. However, the assays shown do not provide the binding affinity, either the text needs to be changed or the data processed to yield binding affinities (as Kd).

2) The assembly of complexes on the membranes from E. coli expressing 3AB is interesting (see Fig 6). It would also have been interesting to complement the studies shown in Fig 5 with assays using the wt 3AB (in membranes) with mutant forms of the 3Dpol protein that had modifications in the different binding regions.

3) It would be good to test whether functional complexes are generated on the bacterial membranes, e.g. by demonstrating uridylylation in the presence of 3CD.

Reviewer #3: Overall, the experimental work is well executed and of high quality, and there is no major need for additional experiments. The text is also well written, but there are some major inconsistencies between Figures 1-3 content and their legends that need to be remedied.

**Part III – Minor Issues: Editorial and Data Presentation Modifications**

Reviewer #1: The authors mention that it was not possible to obtain crystals of 3Dpol with 3B2. Did they perhaps try using a 3B13B2 peptide? See lines 553-4.

Line 81 – what does strictly conserved mean?

Line 106-7 – have the authors checked the sequence database to ensure that this statement is correct?

A schematic figure of the genome showing the 3Bs and 3Dpol would be a useful addition.

It might be useful to include resolution information into the abstract.

Some of the language could be improved in places, for clarity.

The following are examples and not a definitive list:

Line 99 and 523 …unlike the rest of Picornaviruses….unlike other Picornaviruses

Line 100 …all of them were found …all of these were found…

Line 325 3D polymerase…3Dpol

Line 438 On the other hand…However,

Line 501…contained…included

Line 540 How can the fibers be endless?

Line 554 should be FMDV

Reviewer #2: a) Line 433, it seems the text needs changing instead of “reinforce/stress/deeply characterize”

b) There are many minor errors in the use of English that could be corrected by a native English speaker.

c) In the title, “essential” does not really work by itself. It could be “as an essential component to”

d) Line 199 1,5µg/ml should be 1.5µg/ml and I think on line 204 it should be 100 µl/well rather than 100ml/well. See also line 265.

e) Line 215-216, it should be “site-directed mutagenesis”

f) Line 232, the text refers to a western blot but it is not clear what antibody was used for this.

Reviewer #3: Figure 2A is interesting in that it shows an alignment of 3Dpol fibers with intervening 3B1 structures, but is likely somewhat of a misrepresentation of all the packing interactions holding the crystal together. Are there additional polymerase-polymerase contacts in the other dimension of the P212121 and P3231 lattices that contribute to packing? Having only the 3B1 peptide just appears too flexible to yield a high resolution crystal, but perhaps not.

In Figure 3A, consider adding a color-matched indicator (circle?) next to the virus names for the sequences that shown in panels C-G. Or indicate virus names in panels C-G.

PLOS authors have the option to publish the peer review history of their article (what does this mean?). If published, this will include your full peer review and any attached files.

Reviewer #1: No

Reviewer #2: No

Reviewer #3: No

Figure Files:

Data Requirements:

Reproducibility:

References:

---

## [Editor Report · Decision Letter 1]

18 Apr 2023

Dear Prof. Verdaguer,

We are pleased to inform you that your manuscript 'Dual role of the foot-and-mouth disease virus 3B1 protein in the replication complex: as protein primer and as an essential component to recruit 3Dpol to membranes' has been provisionally accepted for publication in PLOS Pathogens.

Best regards,

George A. Belov, PhD

Academic Editor

PLOS Pathogens

Ana Fernandez-Sesma

Section Editor

PLOS Pathogens

Kasturi Haldar

Editor-in-Chief

PLOS Pathogens

orcid.org/0000-0001-5065-158X

Michael Malim

Editor-in-Chief

PLOS Pathogens

orcid.org/0000-0002-7699-2064
---

## [Editor Report · Acceptance letter]

26 Apr 2023

Dear Prof. Verdaguer,

We are delighted to inform you that your manuscript, "Dual role of the foot-and-mouth disease virus 3B1 protein in the replication complex: as protein primer and as an essential component to recruit 3Dpol to membranes," has been formally accepted for publication in PLOS Pathogens.

Best regards,

Kasturi Haldar

Editor-in-Chief

PLOS Pathogens

orcid.org/0000-0001-5065-158X

Michael Malim

Editor-in-Chief

PLOS Pathogens

orcid.org/0000-0002-7699-2064